

# Two-degree of freedom Mahalanobis classifier for smartphone-camera identification from natural digital images

Rubén Vázquez-Medina[1], César Enrique Rojas-López[2],
Omar Jiménez-Ramírez[2], Luis Niño-de-Rvera-Oyarzabal[2] and
Leonardo Palacios-Luengas[3]

[1] Instituto Politécnico Nacional, CICATA Querétaro, Santiago de Querétaro, Querétaro, Mexico
[2] Instituto Politécnico Nacional, ESIME Culhuacan, Ciudad de México, Mexico
[3] Department of Electrical Engineering, Autonomous Metropolitan University, Iztapalapa, Ciudad de México, Mexico

## ABSTRACT

The portability and popularity of smartphones makes it easy to capture digital images in a variety of situations, including witnessing criminal activity. Forensic analysis of digital images captured by smartphone-cameras could be used for legal and investigative purposes, not only to have a recording of an act, but also to establish a relationship between a digital image and its capture device, and between the latter and a person. Fortunately, given the similarities, forensic ballistics techniques and procedures used to identify weapons from fired bullets can be used to identify smartphone-cameras from digital images. However, while there are several solutions for identifying smartphone-cameras from digital images, not all of them focus on two key issues: reducing the number of reference images used to create the fingerprint of the smartphone-camera and reducing the processing time for identification. To address these issues, a method based on a two-degree-of-freedom discriminant analysis using pixel intensity and intrinsic noise in digital images is proposed. It uses a Mahalanobis classifier to compare the traces left by the capture source in a digital image with the fingerprints calculated for the candidate smartphone-cameras. This allows the identification of the most likely smartphone-camera that captured a digital image. A significant advantage of the proposed method is that it relies on a smaller number of reference images to estimate the smartphone-camera fingerprints. They are built using only fifteen reference images, as opposed to thirty or more images required by other techniques. This means faster processing times as image clippings are analyzed rather than whole digital images. The proposed method demonstrates high performance, since for disputed flat images it achieves an identification effectiveness rate of 87.50% with one reference image, and 100.00% when fifteen reference images are considered. For disputed natural images, it achieves an identification effectiveness rate of 97.50% with fifteen reference images.

Corresponding author
Rubén Vázquez-Medina,
ruvazquez@ipn.mx

## INTRODUCTION

The ubiquity of the Internet and the proliferation of smartphones have made it easier to share digital images. Smartphones are personal communication devices equipped with at least one digital camera capable of capturing digital images. The widespread use of smartphones has forced several countries to adapt their legal frameworks to allow digital images to be used as evidence in trials. Similar to ballistics investigations, when analyzing digital images, it is critical to identify the specific camera that captured the image in question (disputed image) in order to establish the identity of the camera owner. From a forensic perspective, it is critical to determine the image-camera and camera-owner links, and an easy way to make these links is through digital image metadata. However, as highlighted in *Sandoval Orozco et al. (2014)*, it is important to note that metadata can be easily and effectively modified or erased. Therefore, alternative approaches rely on traces or patterns left by the smartphone-camera in its captured digital images. These camera traces can be used to create a unique camera fingerprint, which uniquely identifies the smartphone-camera that captured a digital image. It is essential to recall that, as emphasized in *Fridrich (2009)* and *Goljan, Fridrich & Filler (2009)*, a camera fingerprint must meet four critical features: *a) Dimensionality*-It should be finite with a random appearance. *b) Universality*-It should be extractable from all digital images except for black digital images. *c) Permanence*-It should remain unchanged over time and under varying environmental conditions such as weather, temperature, or humidity. *d) Robustness*-It should withstand typical image processing operations, including lossless compression, filtering, and gamma correction. Some authors have demonstrated that the photo response non-uniformity (PRNU) satisfies the aforementioned four key features, making it a valuable component for purposes of producing camera fingerprints. In this context, one of the most significant PRNU-based methods was proposed by *Goljan, Fridrich & Filler (2009)*, which incorporates denoising and Wiener filtering for camera identification from digital images. This approach uses denoising filtering to extract the total image noise and the noise-free image from a noisy image. Wiener filtering is then applied to separate additive noise and multiplicative noise from the total image noise.

Building upon Goljan's method, this work introduces a two-degree-of-freedom discriminant analysis that uses a Mahalanobis classifier to identify smartphone-cameras from disputed images and the controlled capture of some reference images. The Mahalanobis classifier leverages the Mahalanobis distance (MD), calculated between the two-dimensional camera-fingerprint and the two-dimensional traces left in a disputed digital image by its capture device. This calculation aids in determining the most likely smartphone-camera that captured the disputed image. Remarkably, in this study, a smartphone-camera fingerprint is estimated using the camera-in-image traces extracted from a set of reference images captured under controlled conditions by each smartphone-camera under analysis.

As an aside, according to *Suwanda, Syahputra & Zamzami (2020)* and *Estrada (2024)*, it is also important to note that other common metrics such as the Euclidean, Manhattan, and cosine distances, can be used to measure the distance between points in a multivariate

space. However, unlike such common metrics, and as discussed in *McLachlan (1999)*, the Mahalanobis distance considers the correlation between random variables and measures the distance between a point and a distribution in a multivariate space. This is precisely the issue of this study, since the definition of a camera-fingerprint, or the camera-in-image traces left by a camera in its captured images, considers two features of each digital image pixel: its intensity and its PRNU. Therefore, considering that these two features are interrelated, the use of MD is meaningful and allows to distinguish camera-fingerprints with less ambiguity than with the use of other common metrics. On the other hand, in reviewing the state of the art, we found that other researchers have conducted studies on this topic, for example, the Mahalanobis distance has been used as a dissimilarity measure in clustering algorithms for image segmentation by *Zhao, Li & Zhao (2015)* and as a tool for enhancing images of ancient murals by *Nasri & Huang (2022)*. This review of the state of the art led to the idea of using the Mahalanobis distance to identify smartphone-cameras from digital images. Unlike the proposed method, machine learning-based systems require large amounts of data to train the models, as well as significant development and maintenance resources and costs.

## Related works

Mahalanobis classifiers are Gaussian classifiers that rely heavily on the MD as their underlying distance method. Despite its widespread use, the MD may experience a loss of performance when considering a multidimensional approach in the presence of noise. This is due to the equally weighted sum of squared distances over all features. As a result, features with the greatest distance may dominate and mask all other features. Hence, only the dominant features are considered in a classification process, and the information provided by the remaining features is neglected. However, this effect seems to be less relevant in this study, since the interesting features in the objects of study represent noise signals. Mahalanobis classifiers find applications in various domains. For instance, *Babiloni et al. (2001)* used them to recognize electroencephalography (EEG) patterns with the advantage of using a limited number of EEG electrodes. In the same year, *Imai, Tsumura & Miyake (2001)* used the MD to estimate perceptual color differences in digital images, especially when a transformation such as gamut-mapping is required. *Zhao, Li & Zhao (2015)* used the MD as a dissimilarity measure in clustering algorithms for image segmentation. *Srivastava & Rao (2016)* proposed an MD-based algorithm to classify text from very large correlated datasets. On the other hand, *Siddappa & Kampalappa (2020)* used an MD-based learning method in the nearest neighbor (NN) to improve the classification performance in the unbalanced dataset. Additionally, *Li et al. (2020)* proposed an MD-based algorithm to classify and detect changes in the distribution of dataset over time. *Raiyani et al. (2022)* used the MD to estimate misclassification caused by K-nearest neighbors (KNN), extra trees (ET), and convolutional neural network (CNN) models when classifying Sentinel-2 images into six scene classes. *Nasri & Huang (2022)* used the MD to improve images of ancient murals by extracting specific degraded segments from red-green-blue (RGB) images. *Wang, Kang & Zhang (2023)* proposed an MD-based deep domain adaptation model for self-supervised attention correlation

alignment that learns representations useful for solving other subsequent tasks of interest but does not require label data. Also, *Warchoł & Kapuściński (2023)* proposed an MD-based method for hand posture recognition using skeletal data, which includes a two-stage classification algorithm with two additional stages related to joint preprocessing, and a rule-based system. *Ghosh et al. (2024)* proposed a semi-parametric MD-based classifier for observing competing classes, which estimates the posterior probability of each class. Finally, in the work developed by *Gioia (2024)* an R package can be used to compute the MD between each pair of species in data frames, considering that each data frame contains the observations of one species with some variables. The R package, named "cmahalanobis", uses the statistics functions of R for matrix computation.

The topic of smartphone-camera identification from digital images is burgeoning in the field of forensic science and attracting numerous researchers. For instance, *Bernacki (2020)* compiled and analyzed research efforts from 2010 to 2020, providing an overview of methods and algorithms for digital camera identification, focusing on sensor identification as a classification key strategy. According to *Bernacki (2020)*, the camera-in-image traces (traces left by the digital camera on each captured image) are unique artifacts that can be used to define the camera-fingerprints useful in passive forensic analysis techniques. Likewise, *Nwokeji et al. (2024)* presented a survey of all existing methods and techniques for identifying digital cameras using both intrinsic hardware artifacts, such as sensor pattern noise and lens optical distortion, and software artifacts, such as color filter array and automatic white balance. In this survey, *Nwokeji et al. (2024)* reviewed the existing capture source identification methods, their evaluation criteria, and the publicly available datasets used to evaluate their performance. Building upon this, *Zeng et al. (2019)* developed a PRNU-based method for identifying smartphone-cameras in zoomed-in digital images, a common output of these devices. Additionally, *Freire-Obregón et al. (2019)* proposed a method for identifying digital cameras using deep learning and the noise pattern of the smartphone-camera sensors. Similarly, *Berdich & Groza (2022)* proposed a method for identifying smartphone-cameras by considering camera sensor features, which was based on the low-mid frequency coefficients of the discrete cosine transform (DCT) on the dark signal non-uniformity (DSNU). Furthermore, *Rodríguez-Santos et al. (2022)* proposed a method for identifying smartphone-cameras from natural images. It was based on residual noise and the Jensen-Shannon divergence. This approach can be seen as an extension of the study presented by *Quintanar-Reséndiz et al. (2021)*, which used flat images and the Kullback-Leibler divergence in the identification process. On the other hand, *Qian et al. (2023)* presented a capture source identification framework that uses sensor noise enhanced by neural networks to trace back web photos while providing cryptographic security. *Manisha, Li & Kotegar (2023)* proposed a capture source identification technique by defining a global stochastic fingerprint from the low- and mid-frequency bands of digital images and evaluating its resistance to perturbations in the high-frequency bands. They concluded that their technique could potentially be used to identify video-based capture sources. *Berdich, Groza & Mayrhofer (2023)* extended device authentication beyond the smartphone-camera to the smartphone itself by reviewing existing approaches that define smartphone fingerprints based on internal components

such as camera sensors, speakers, microphones, accelerometers, magnetometers, gyroscopes, and radio frequency sensors. In their study, *Berdich, Groza & Mayrhofer (2023)* also provided an overview of the most common feature extraction techniques for all types of sensors. Specifically, they found that PRNU, DSNU, fixed pattern noise (FPN), local binary pattern (LBP), being processed in the DCT, spectral, or spatial domain, were used by several authors to generate camera-fingerprints identifying the camera sensors. In addition, *Shuwandy et al. (2024)* conducted a systematic review of camera sensor-based smartphone authentication methods and discussed current challenges and issues by analyzing studies reported in four digital scientific databases: ScienceDirect, IEEE Xplore, Web of Science, and Scopus. In video forensics, *Anmol & Sitara (2024)* used PRNU and a classifier based on a support vector machine (SVM) to verify the authenticity of the videos. On the assumption that PRNU can be affected by highly textured content or post-processing, they combined the PRNU higher order wavelet statistical information of a video I-frame with the information provided by the LBP and the gray level co-occurrence matrix (GLCM).

As far as can be seen from the review of the specialized works reported in this section, it can be noted that the published methods for the identification of digital cameras do not include Mahalanobis classifiers, as proposed in this work.

## Main contribution

The main contribution of this work is a method for identifying smartphone-cameras from both flat and natural digital images. The proposed method encompasses the following key features:

- A two-degree-of-freedom classification system that uses the Mahalanobis distance between two-dimensional variables that are assumed statistical instead of one-dimensional correlation functions.
- Instead of relying solely on PRNU, this method uses two-dimensional features that combine PRNU, and pixel intensity extracted from digital images.
- For defining the two-dimensional fingerprint of smartphone-cameras, this method uses an approach that, instead of processing entire images and utilizing thirty or more reference images, extracts $1,000 \times 1,000$-clippings from fifteen reference images. This significantly reduces the data requirements and consequently the processing time during camera identification.
- The proposed method relies on the Mahalanobis distance to compare the two-dimensional camera fingerprints pre-estimated for the suspected smartphone with the camera-in-image traces.
- This method can distinguish between even identical smartphone-cameras (same brand and model) from both flat and natural images.
- The effectiveness of the proposed method was greater than 97% when using fifteen reference images to build the smartphone-camera fingerprints.

This work is organized as follows. Section "Image Dataset and Intrinsic Noise" describes the image dataset used to prepare the case studies and the method of extracting noise from a digital image according to the noise model proposed by *Goljan, Fridrich & Filler (2009)*. Section "Approaches to digital camera identification" demonstrates how to apply discriminant analysis using two approaches. The first one uses the probability distribution function (PDF) for the pixel intensity from the green layer of the available digital images. This is done because the camera sensor has three types of color filters: green, red and blue filters, and for every blue and red filter there are two green filters. Considering that the color map was designed to mimic the human retina, it is assumed that there is more information in the green layer of the digital image than in the other two layers. In addition, the green layer is particularly useful for analyzing the luminance of the image and detecting any compression or noise problems in the RGB channels. The second one considers a discriminant plane consisting of pixel intensity and PRNU extracted from the green layer of the flat images for each smartphone-camera. It is also demonstrated that by defining smartphone-camera fingerprints in this way, it is possible to distinguish smartphone-camera without any ambiguity. From essential concepts, the section "Proposed Method" presents the proposed classification method, which is based on the second approach of the previous section to define smartphone-camera fingerprints distinguished by Mahalanobis distance. Section "Case Studies" presents two case studies and how they were prepared, specifying the disputed images and smartphone-cameras features considered in them. Section "Discussion and Comparison with other Methods" shows a comparison between the proposed method and other similar methods based on PRNU and some divergence measures. In this section, a comparison table summarizes the features of the methods in terms of image dataset, number of images, number of devices, identification strategy, and identification rate. Finally, the conclusions are presented.

## IMAGE DATASET AND INTRINSIC NOISE

### IPN-NFID: flat and natural image dataset

In this work, the IPN-NFID image dataset developed by *Rojas-López et al. (2024)* available at https://doi.org/10.6084/m9.figshare.25201319 was considered, which currently contains digital images captured by twelve smartphone-cameras. This dataset contains twelve hundred flat images, six hundred landscape natural images, and three hundred and sixty portrait natural images. For the purposes of this work, it is assumed that a flat image has a neutral appearance (approximately gray) and low contrast, resulting in weak details in highlights and shadows. Furthermore, a gray surface was used to generate the reference images, whose plane was at a perpendicular angle to the capture direction. On the other hand, a natural image includes everything that a human would observe in the real world, including scenes with contrast and intensity such as roads, mountains, beaches, people, objects, and animals, among others. For this work, as explained in the "Case studies" section, only the eight smartphone-cameras that originally comprised the IPN-NFID image dataset were considered. Over time, this image dataset has been enhanced, and by the end of this project, it includes flat and natural digital images from twelve smartphone-cameras. Table 1 shows the initial configuration of the IPN-NFID image dataset.

**Table 1 Smartphone brands and models as well as the number of flat and natural images that originally constituted the image dataset used for the case studies.**

| Device id | Brand | Model | Portrait flat images | Landscape natural images | Portrait natural images | Dimensions |
|---|---|---|---|---|---|---|
| $C_{01}$ | iPhone | SE 2020-1 | 100 | 50 | 30 | 4,032 × 3,024 |
| $C_{02}$ | iPhone | XR | 100 | 50 | 30 | 4,032 × 3,024 |
| $C_{03}$ | Motorola | G4 Plus | 100 | 50 | 30 | 4,608 × 3,456 |
| $C_{04}$ | Samsung | Galaxy A01 | 100 | 50 | 30 | 4,160 × 3,120 |
| $C_{05}$ | Samsung | Galaxy Note 9 | 100 | 50 | 30 | 4,032 × 3,024 |
| $C_{06}$ | Motorola | G20 | 100 | 50 | 30 | 3,000 × 4,000 |
| $C_{07}$ | iPhone | SE 2020-2 | 100 | 50 | 30 | 4,032 × 3,024 |
| $C_{08}$ | Huawei | Y9 2019 | 100 | 50 | 30 | 4,160 × 3,120 |

It is important to note that the flat images were taken with a distance of fifteen centimeters between the smartphone-camera and the reference color paper in a room with controlled lighting, ensuring consistent illumination for all shots. Meanwhile, natural images were taken under the lighting conditions of the location and time of the shoot day and at an arbitrary distance from the target scene.

In order to identify the capture source of a set of disputed images, the experiments conducted in the "Case studies" section of this work used fifteen instead thirty flat images as reference images per smartphone-camera to define each camera fingerprint. Note that these experiments demonstrated that a smaller number of reference images per camera can be used to define the camera-fingerprint of each smartphone under consideration compared to other techniques. Additionally, ten flat or natural images (from 41-th to 50-th image) per smartphone-camera were selected as disputed images from the used dataset. It is also considered that the pixel intensity and PRNU matrices extracted from each disputed image contain information about the camera-in-image traces left by its capture source. It is important to emphasize that when considering disputed flat images, a 1,000 × 1,000 clipping was extracted from each image to obtain the pixel intensity and PRNU matrices. However, in the case of the natural disputed images, these matrices were obtained from a pixel set taken from each image using a pixel selection strategy based on the pixel intensity of flat images capturing gray reference surfaces. In this way, as demonstrated in the "Case studies" section, the method proposed in this work is faster than other methods because it considers a part of the image to be analyzed instead of the whole image.

## Intrinsic noise in digital images

This study adopts the noise model for a digital image shown in Eq. (1). In this model, $i(x, y)$ represents the pixel intensity at coordinate $(x, y)$ of the green layer of a noisy image, $i_0(x, y)$ represents the corresponding pixel intensity at coordinate $(x, y)$ on the free-noise image related to $i(x, y)$, $\eta_m(x, y)$ signifies the multiplicative noise, and $\eta_a(x, y)$ is the additive noise from $i(x, y)$, both at the same coordinate $(x, y)$.

$$i(x, y) = i_0(x, y) + \eta_m(x, y)i_0(x, y) + \eta_a(x, y). \tag{1}$$

Similar to *Goljan, Fridrich & Filler (2009)*, according to Eq. (2), some denoising filtering technique can be applied to $i(x, y)$.

$$i_0(x, y) \approx f_d[i(x, y)]. \tag{2}$$

It is worth noting that $f_d[i(x, y)]$ decomposes $i(x, y)$ into a coefficient series across over frequency bands, and it uses a suitably chosen threshold for each band, setting to zero the coefficients that most influence the image noise. It should be noted that the noise is affected by the small coefficients, while the pixel intensity is affected by the large coefficients. Although the main purpose of denoising filtering is to eliminate or minimize the effects of unwanted artifacts embedded in the signals of interest, preserving them with the highest possible quality, in this case it is used to recover from a digital image the artifacts left by its smartphone-camera, while preserving the contextual details of the original images. The most commonly used denoising filters are mean, median, Gaussian, bilateral, and wavelet. Some of these can be applied in both the spatial and spectral domains, while others can only be applied in one of the two domains. For further details, the work proposed by *Xiao & Zhang (2011)* should be consulted.

In this way, from Eq. (1), the residual noise, $\eta(x, y, i_0)$, from a digital image can be estimated by Eq. (3), in which $\eta_m(x, y)$ and $\eta_a(x, y)$ are specified.

$$\eta(x, y, i_0) \approx i(x, y) - f_d[i(x, y)] = \eta_m(x, y)i_0(x, y) + \eta_a(x, y). \tag{3}$$

Now, following the approach of *Jingdong et al. (2006)*, the Wiener filtering, $f_w[.]$, is applied to Eq. (3) to obtain the PRNU of any digital image, $prnu(x, y) = \eta_m(x, y)i_0(x, y)$ must be computed for all coordinates $(x, y)$ in the digital image according to Eq. (4).

$$prnu(x, y) = f_w[i(x, y) - f_d[i(x, y)]] = \eta_m(x, y)i_0(x, y). \tag{4}$$

Therefore, it should be noted that Wiener filtering techniques have been used to eliminate or reduce the additive noise in the digital images, which is assumed as a stationary random process that does not depend on the pixel location. More specifically, based on Eq. (4), Wiener filtering is used in this case to remove the additive component of the camera-in-image traces left in each pixel by the capture source. In general, these filtering techniques are designed to minimize the quadratic error between an image considered as the original image and another image considered as the reconstructed image. These filtering techniques are variable cutoff low-pass, using a lower cutoff frequency for low-detail regions and a higher cutoff frequency for regions with edges or high variance features. For more details, see the works proposed by *Vaseghi (1996)* and *Khireddine, Benmahammed & Puech (2007)*.

On the other hand, it should be highlighted that PRNU depends on lighting and is caused by small differences in pixel intensity due to the camera sensor, resulting in a brighter or darker pixel effect in the image. This is consistent with Eq. (4) since the pixel intensity is increased (making the pixel brighter) when PRNU is positive or decreased

(making the pixel darker) when PRNU is negative. It is worth noting that PRNU is not affected by camera sensor temperature and exposure time during digital image capture. PRNU then quantifies the influence of the camera sensor on the intensity of each image pixel, affecting its brightness (or darkness) depending on whether it is positive or negative.

Finally, note that according to Eq. (4), $prnu(x, y)$ depends on $i_0(x, y)$, so if $i_0(x, y)$ changes in its magnitude and statistical behavior, it is expected that $prnu(x, y)$ must also change in its magnitude and statistical behavior. Therefore, by evaluating Eqs. (2) and (4) for all pixels of the digital image $r$ captured by the camera of the $j$-th smartphone, the pixel intensity and PRNU matrices for $r$ can be identified as $I_{0,[r,j]}$ and $PRNU_{[r,j]}$, respectively.

## APPROACHES TO DIGITAL CAMERA IDENTIFICATION

### An approach to the issue of using natural images

The main problem with trying to distinguish digital cameras from natural images is that their scene contextual information significantly distorts the camera-in-image traces. Therefore, the method for distinguishing between digital cameras is less effective using natural images than using flat images as disputed images. One approach to address this problem is to study and propose strategies to generate a flat equivalent image derived from the natural image under analysis. It is important to emphasize that the statistical behavior of pixel intensity of these flat equivalent images should be close to that of flat images captured under controlled conditions by the same smartphone-camera. This strategy can reduce the impact of contextual information in natural images on the effectiveness of methods for distinguishing digital cameras, since only natural image pixels whose intensity is close to that of flat image pixels are processed, and the remaining pixels in a natural image are ignored. Similar to *Quintanar-Reséndiz et al. (2021, 2022)*, and *Rodríguez-Santos et al. (2022)*, the calculation of the flat equivalent image for a natural image was based only on the information extracted from its green layer.

The plots in Fig. 1 illustrate some examples that demonstrate the statistical behavior of $I_0$ extracted from the green layer of digital images, allowing to distinguish the capture source of the digital image under analysis. Figure 1A shows the computed PDF of $I_0$ extracted from a flat image for each of the available smartphone-cameras, whereas Fig. 1B shows the PDF computed from the average of $I_0$ with fifteen flat images. Note that although the smartphone-cameras captured the same surface under similar lighting conditions, the statistical behavior of pixel intensity of the captured digital images could be used to distinguish them, since the computed PDFs for $I_0$ changed significantly. Note also, in Figs. 1A and 1B that both sets of PDFs for $I_0$ exhibit a slight potential for distinguishing smartphone-cameras. However, they may lead to confusion when dealing with highly similar PDFs, such as those for twin smartphone-cameras.

On the other hand, note in Fig. 2A that the discrimination process can be inefficient when the PDF computed from the average of $I_0$ using fifteen flat images is compared to a PDF computed for $I_0$ with a single flat image. Furthermore, note in Fig. 2B that the inefficiency of the discrimination becomes critical when the PDF computed from the average of $I_0$ using fifteen flat images is compared to the PDF computed for $I_0$ of a natural image. It is worth noting that Fig. 2B demonstrates that the statistical behavior of $I_0$

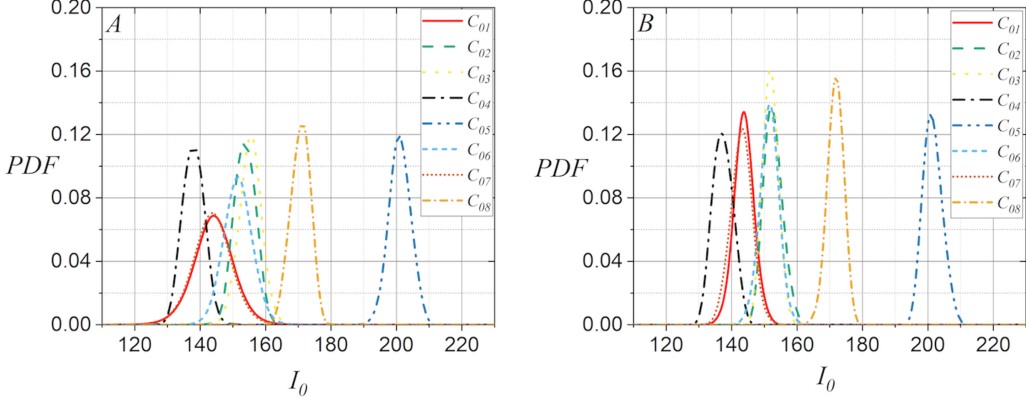

**Figure 1** PDF for $I_0$ extracted from the green layer of images captured with the smartphone-cameras used in the case studies: (A) Using one flat image per camera and (B) using fifteen flat images per smartphone-camera.

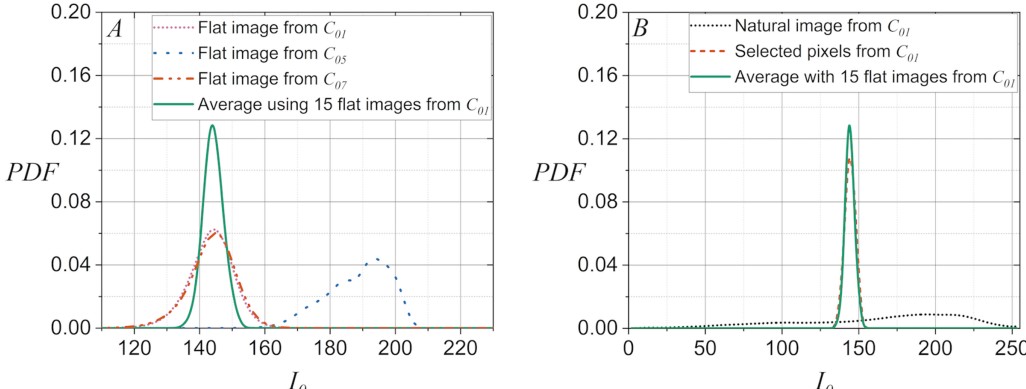

**Figure 2** PDFs for $I_0$ extracted from the green layer of images captured with the smartphone-cameras used in the case studies: (A) Using a flat image in $C_{01}$, $C_{07}$, and $C_{05}$, and using the average of $I_0$ with fifteen flat images in $C_{01}$, (B) for $C_{01}$ a natural image and for the selected pixels of same image, and the average of $I_0$ with fifteen flat images.

extracted from a natural image can be matched to that of a flat image. This is done by performing a pixel selection on the natural images to create the flat equivalent image with a PDF computed for $I_0$ similar to that of a flat image. This latter idea allows smartphone camera identification methods, which are efficient when using flat images, can be applied to cases where the disputed images are natural images.

Considering this analysis, and using the model established by Eq. (4), it can be expected that $I_0$ extracted from an image under analysis affects its PRNU. This fact is considered in the following section.

## An approach using two-dimensional variables

From previous results and based on the model given in Eq. (4), it is considered a smartphone-camera identification approach that involves two-dimensional variables on a Bayesian classifier. This approach considers the discriminant plane showed in Fig. 3, which allows to identify that $I_0$ relates to PRNU when they are extracted from green layer of the

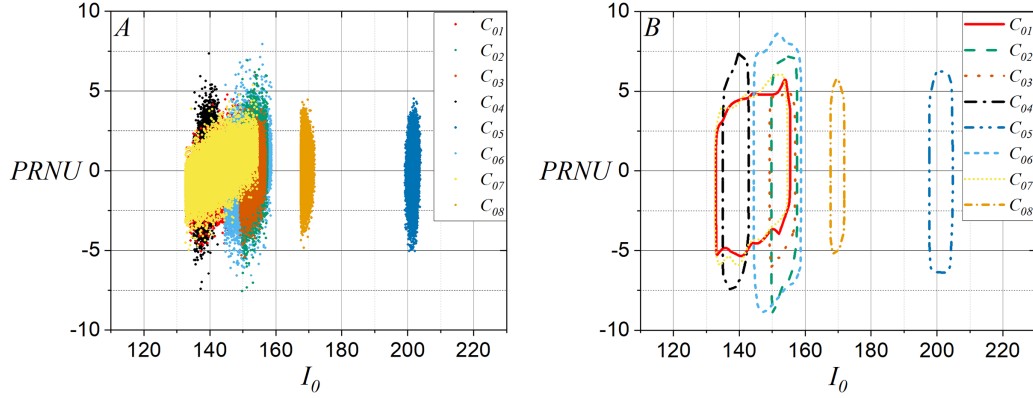

**Figure 3 Discriminant planes $I_0$ vs. PRNU using a flat image per smartphone-camera: (A) All points in the flat images and (B) contours for (A).**

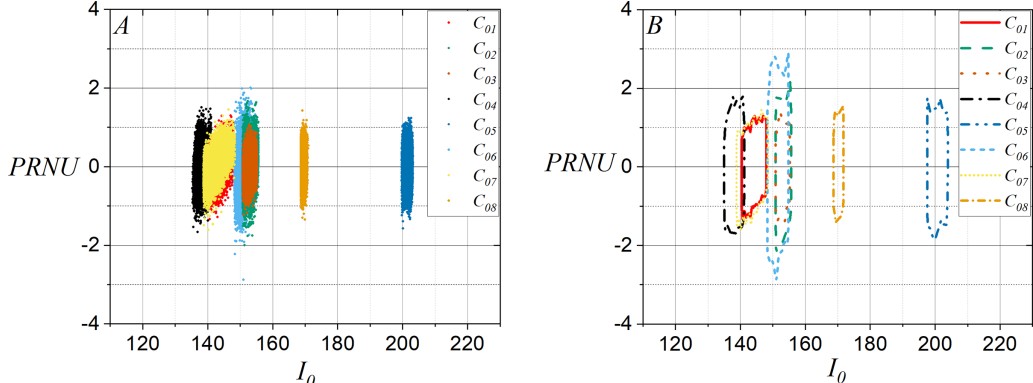

**Figure 4 Discriminant planes for the average computed of $I_0$ and PRNU using fifteen flat images per smartphone-camera: (A) All points in the average of $I_0$ and PRNU and (B) contours for (A).**

flat images used to estimate the smartphone-camera fingerprints. Note in Fig. 3A that the model proposed by Eq. (4) is consistent with the fact that if $I_0$ changes then PRNU also changes when using one flat image. Figure 3B shows the contours of the plots in Fig. 3A.

Similarly, it should be noted in Fig. 4A that the discriminant planes were defined for $I_0$ and PRNU using fifteen flat images, and Fig. 4B shows the contours of plots in Fig. 4A. Note that the plots have become sharper and more compact than shown in Fig. 3, which can increase the effectiveness of distinguishing one smartphone-camera from another.

Additionally, Fig. 5A shows for $C_{01}$ the contours taken of the discriminant planes for $I_0$ and PRNU using fifteen flat images, and it shows the contours for $C_{01}$, $C_{05}$, and $C_{07}$ when a flat image is used. It can be observed that in the discriminant plane, $C_{01}$ and $C_{07}$ have contours similar to the average contour for $I_0$ and PRNU using fifteen planar images. On the other hand, the contour for $C_{05}$ is very different from all the others. This is due to the fact that $C_{01}$ and $C_{07}$ have the same brand and model. Figure 5B shows the contours in the discriminant plane for $I_0$ and PRNU when one disputed natural image is taken from $C_{01}$ with and without pixel selection to be compared against the contour of the discriminant

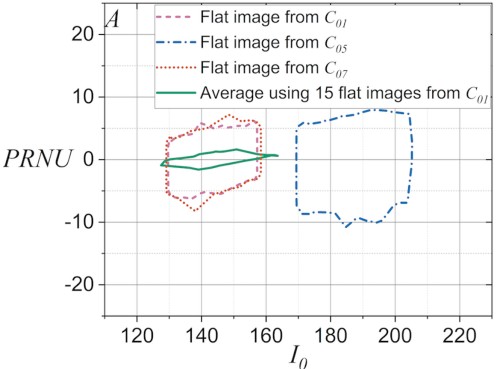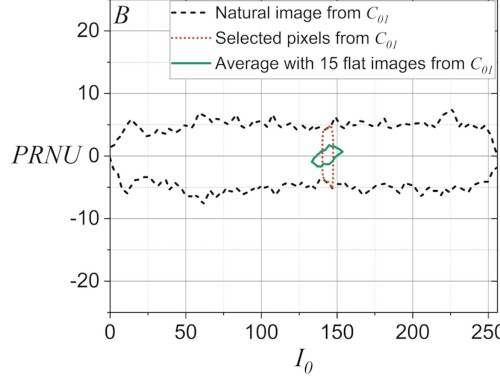

**Figure 5 Discriminant plane considering the average computed for $I_0$ and PRNU with fifteen flat images, and its comparison with the discriminant plane for three smartphone-cameras.** (A) Disputed flat images from $C_{01}$, $C_{05}$, and $C_{07}$, and for the average computed for $I_0$ and PRNU with fifteen flat images from $C_{01}$ and (B) a disputed natural image with and without pixel selection, and the average computed for $I_0$ and PRNU with fifteen flat images from $C_{01}$.

plane for the average for $I_0$ and PRNU with fifteen flat images taken with $C_{01}$. Note in Fig. 3 through Fig. 5 that the colors identify each smartphone-camera considering its identifier given in Table 1. It should be emphasized that the two-dimensional variables in the discriminant plane are useful when the disputed images are flat images. However, without pixel selection, it is not useful when the disputed images are natural images, as the scene content influences it, preventing a clear relation to the two-dimensional variable of the reference image. Note in Fig. 5B that the contour of the two-dimensional variable for the natural image of $C_{01}$ is very wide. Then, this situation suggests selecting the pixels in the natural image whose intensity is closest to the intensity of the pixels contained in the reference image.

The experiments presented with both approaches help to formulate the method proposed in this study. On the one hand, a flat equivalent image computed from a natural image was generated by applying a pixel selection strategy. Also, a comparison strategy for distinguishing smartphone-cameras was developed that relies on two-dimensional variables whose components exhibit a dependency between them.

## PROPOSED METHOD

### Essential concepts

The method proposed in this study uses two-dimensional variables extracted from digital images. It is based on the camera-in-image trace, the smartphone-camera fingerprint, and the Mahalanobis distance. These three essential concepts are described below.

- **Camera-in-image traces**. It is assumed that the traces left by the $q$-th smartphone-camera in the digital image $d$ can be observed using a two-dimensional variable consisting of $I_{0,[d,q]}$ and $PRNU_{[d,q]}$ that have some relationship to each other. When the digital image $d$, flat or natural, captured by the camera of the $q$-th smartphone, is analyzed in a trial, the two-dimensional variable extracted from its green layer containing these traces is defined by $T_{[d,q]} = [I_{0,[d,q]}, PRNU_{[d,q]}]$.

- **Smartphone-camera fingerprints**. The camera fingerprint of the $j$-smartphone is generated from the camera-in-image traces extracted from the $1{,}000 \times 1{,}000$ clippings of the green layer of $R$ reference flat images captured under controlled conditions by the camera of that smartphone. Then, the smartphone-camera fingerprint defined for the $j$-th smartphone is $F_j^R = \frac{1}{R}\sum_{r=1}^{R} T_{[r,j]}$. The $1{,}000 \times 1{,}000$ clippings of the green layer are taken from the center $(cent_x, cent_y)$ in each reference flat image, where $x \in (cent_x - 500,\ cent_x + 500)$ and $y \in (cent_y - 500,\ cent_y + 500)$.

- **Mahalanobis distance**. The proposed classifier is based on the Mahalanobis distance, $MD(s_1, s_2)$, introduced in 1936 by P. Ch. Mahalanobis to determine the similarity of two multidimensional random variables, $s_1$ and $s_2$. Note that the distance between each pair of points in two collections is distributed in a multidimensional space. In addition, it is worth noting that $MD(s_1, s_2)$ differs from Euclidean distance because it takes into account the correlation between multidimensional random variables. Note also that, under the assumption of three random variables $s_1$, $s_2$, and $s_3$, $MD(s_1, s_2)$ satisfies the following conditions: **(a)** *Non-negativity*: $MD(s_1, s_2) > 0$, **(b)** *Symmetry*: $MD(s_1, s_2) = MD(s_2, s_1)$, and **(c)** *Triangular inequality*: $MD(s_1, s_2) \leq MD(s_1, s_3) + MD(s_3, s_2)$. It not only provides a measure for the average distance of two random variables but also measures the correlation between those variables based on their covariance. It can be calculated using Eq. (5).

$$MD(s_1, s_2) = \sqrt[2]{(s_1 - s_2)^T \Sigma^{-1} (s_1 - s_2)}, \tag{5}$$

where $\Sigma^{-1}$ represents the covariance matrix between $s_1$ and $s_2$.

$MD(s_1, s_2)$ is a crucial tool in a Mahalanobis classifier, which is a special case of a Bayes classifier that satisfies the conditions for a linear discriminant analysis (LDA) approach when $s_1$ and $s_2$ are $n$-dimensional features of the class $w_u$, where $u = 1, 2, 3, \ldots$, $p(s_v | w_u)$ obeys Gaussian probability distributions, $p(w_u)$ is constant, and all constant terms have been removed from the comparison between classes.

## Method description

The Mahalanobis classifier proposed in this work is developed considering a passive strategy for the identification of smartphone-cameras from digital images, since it is not required to add any external signal to the digital images under analysis. It consists of the following steps.

i. **Estimation of smartphone-camera fingerprints**. Fifteen reference flat images must be considered to define the camera-fingerprint of each smartphone. Thereby,
$F_j^R = \frac{1}{R}\sum_{r=1}^{R} T_{[r,j]}$, with $R = 15$, $j = 1, 2, 3, \ldots, C$, and $C = 8$.

ii. **Estimation of camera-in-image traces**. Considering $D = 10$ disputed digital images for each smartphone-camera, two options are considered to estimate the traces left by the camera of the $q$-th smartphone in each image $d$, which can be observed by using
$T_{[d,q]} = [I_{0,[d,q]}, PRNU_{[d,q]}]$, with $q = 1, 2, 3, \ldots, C$.

(a) *When the disputed image is a flat image:* $T_{[d,q]}$ is calculated from a $1,000 \times 1,000$ clipping extracted from the green layer in the *d*-th *disputed flat image* of the camera of the *q*-th smartphone.

(b) *When the disputed image is a natural image*: $T_{[d,q]}$ is calculated from the pixels selected from the green layer in the *d*-th whole *disputed natural image* captured with the camera of the *q*-th smartphone. The pixel selection criterion is based on extracting the pixels whose intensity is in $(\mu - 2\sigma, \mu + 2\sigma)$, where $\mu$ is the mean and $\sigma$ is the standard deviation of the pixel intensity vector taken from the $F_j^R$ when the compared camera is from the *j*-th smartphone.

iii. **Calculation of Mahalanobis distance**. According to Eq. (5), the Mahalanobis distance between the smartphone-camera fingerprint and the camera-in-image traces associated to the pixel intensity of a disputed image is calculated using code developed in MATLAB for this study. That is, $MD(T_{[d,q]}, F_j^R)$ for $j = 1, 2, 3,..., C, q = 1, 2, 3,..., C$, and $d = 1, 2, 3,..., D$.

iv. **Comparison of Mahalanobis distances**. The comparison of the calculated Mahalanobis distances is performed according to Eq. (6).

$$MD(T_{[d,q]}, F_q^R) \leq MD(T_{[d,q]}, F_j^R) \,, \tag{6}$$

for $j = 1, 2, 3,..., C, q = 1, 2, 3,..., C$, and $d = 1, 2, 3,..., D$, with the understanding that the first index change in the comparison occurs at *j*, then at *q*, and finally at *d*.

v. **Associate a smartphone-camera to the disputed image**. Associate the camera of the *q*-th smartphone with the disputed image *d* when *q* is the camera that produces the smallest $MD(T_{[d,q]}, F_j^R)$.

Note that for (*i*) and (*ii*) in the proposed method, it is assumed that PRNU has been calculated using Eq. (4), where the applied denoising filter, $f_d[i(x,y)]$, uses a wavelet approach with `wavelet_levels = 4` and `wavelet = (Daubechies, 8)`, and the Wiener filter is based on the two-dimensional Discrete Fourier Transform (DFT). The code proposed by *Goljan, Fridrich & Filler (2009)* was used to implement these filters. The function `NoiseExtractFromImage()` was used for the denoising filter and the function `WienerInDFT()` was used for the Wiener filter supported by the functions `std2`, `fft2`, and `ifft2` from MATLAB. It is important to note that the proposed method uses clippings extracted from the green layer of each reference digital image, and clippings in disputed flat images or selected pixel in disputed natural images. These conditions significantly reduce the processing time and memory requirements of the computer used. The choice of the green layer is not arbitrary, it is selected because it contains twice as much noise information as the other color layers. Figure 6 provides a graphical representation of the proposed method.

## CASE STUDIES

Firstly, it is important to emphasize that the digital images used in the case studies of this study have not been contaminated with any external noise signal, since the aim is to

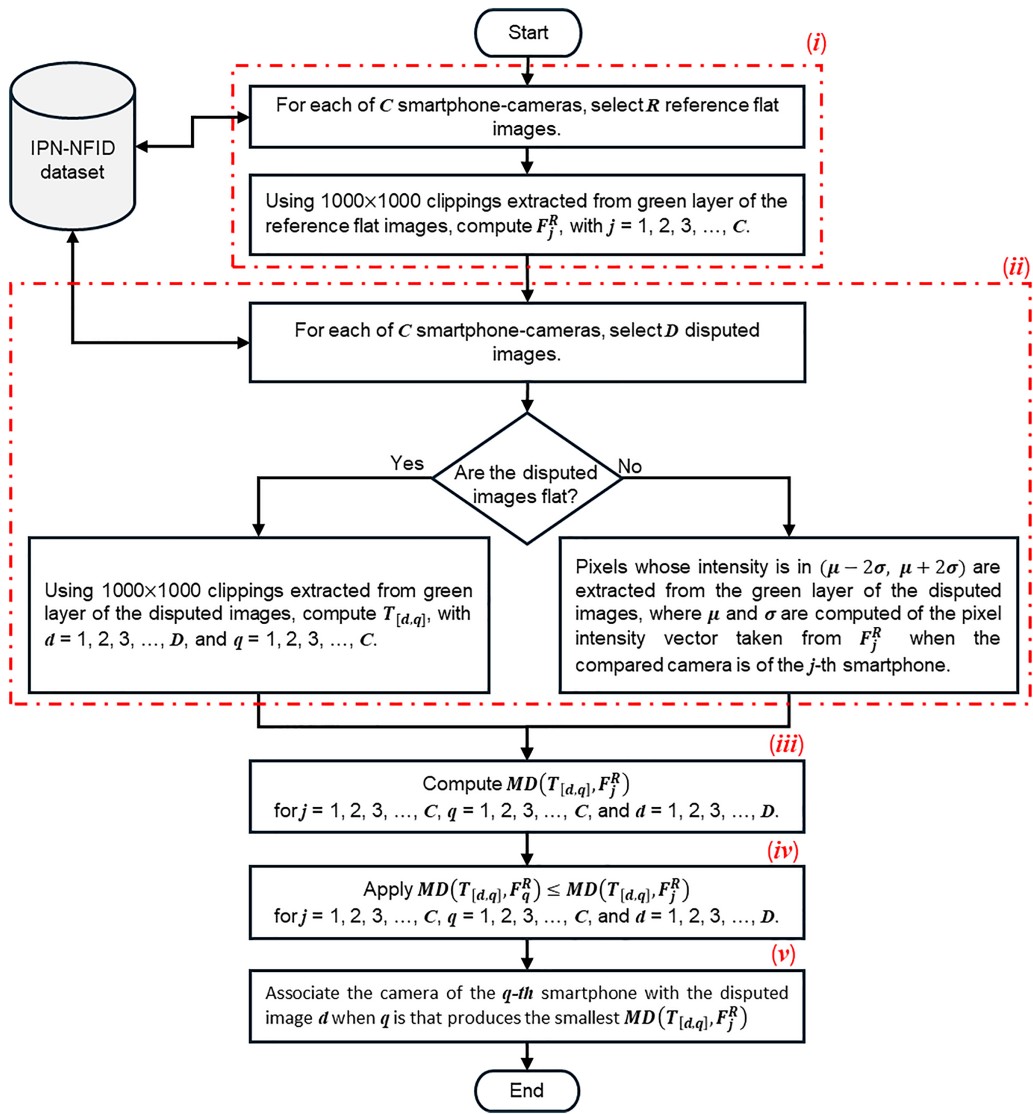

**Figure 6 Flowchart that gets the statistical fingerprint of a smartphone-camera and compares it to the fingerprint of a digital image, flat or natural, to identify its source of capture.**

identify, through a passive method, the capture source of the digital images to be analyzed. In this context, and before proposing and preparing the case studies, the conformation of the dataset was analyzed considering the size of the digital images generated by the smartphone-cameras. On this basis, three groups of smartphone-cameras were identified. The first group consisted of seven smartphone-cameras that generated digital images of approximately 12 megapixels (see Table 1). The second group consisted of three smartphone-cameras that generated digital images of less than 8.6 megapixels. Finally, the third group consisted of two smartphone-cameras that generated digital images of less than 16 megapixels. Using this information the first group was selected, because in it the smartphone-cameras could be confused considering that they produced digital images with similar size. The other groups were excluded because they had fewer smartphone-

cameras and because there was a clear difference in the size of the digital images they produced compared to the group with more smartphone-cameras. Note from a basic analysis shows that the smartphone-cameras in the second group produce digital images that are 31% smaller than the average size of the digital images produced by the smartphone-cameras in the first group. In addition, the smartphone-cameras in the third group produce digital images that are 29% larger than the average size of the images produced by the cameras in the first group. Therefore, eight smartphone-cameras were selected (seven from the first group and one from the third group), and four smartphone-cameras were discarded (two from the second group and two from the third group), because they produced digital images whose size is very different from the average size of the digital images produced by the selected smartphone-cameras. Nevertheless, it was considered appropriate to include in the case studies a smartphone-camera from the third group to test-run the proposed method. On the other hand, the number of smartphone-cameras used in this study was consistent with the number of devices used in other works reported in the peer-reviewed literature (see the "Discussion and comparison with other methods" section). It would be very interesting to perform further experiments with a larger number of smartphone-cameras, including a wider variety of brands and models. However, this action is not necessary to ensure the effectiveness of the proposed method, since in a real forensic case, a reduced number of digital cameras are involved to be analyzed in the identification of the capture source. On the other hand, it is not necessary to use a much larger number of reference images to compute the smartphone-camera fingerprint, since, as demonstrated by *Quintanar-Reséndiz et al. (2021)*, twenty reference images per camera is an acceptable number for capture source identification when using one-dimensional variables. With this baseline in mind, and assuming that camera-in-image traces are registered in two-dimensional variables, fifteen reference images were used in this study to build the smartphone-camera fingerprints. Finally, it is emphasized that a real forensic case should associate a limited number of smartphone-cameras with a limited number of images to be analyzed.

Thus, based on Table 1, two case studies were prepared to demonstrate the effectiveness of the developed method. For case study A, ten flat digital images per smartphone-camera were selected as disputed images. For case study B, ten natural images were selected as disputed images. In both cases, the disputed images were labeled as $d_{i,j}$ where $i$ is the disputed image considered from the camera of the $j$-th smartphone, with $i = 41, 42, 43, \ldots, 50$ and $j = 1, 2, 3, \ldots, 8$. Note, that Table 1 contains two smartphones of the same brand and model: iPhone SE 2020-1 and iPhone SE 2020-2. The digital images of the eight smartphone-cameras in this study were taken from the IPN-NFID image dataset developed by *Rojas-López et al. (2024)* and available at https://doi.org/10.6084/m9.figshare.25201319. For case study A, two approaches were used to define the smartphone-camera fingerprints. The first approach was preliminary and was used to describe in detail the actions performed in the proposed method. In this approach, a single flat image was used to estimate the smartphone-camera fingerprints. The second approach was the one mentioned in (i) of the description of the proposed method. In this approach, the average pixel intensity and PRNU calculated with fifteen flat images were used to estimate each

smartphone-camera fingerprint. According to the description of the proposed method, for case study B, the second approach of case study A was considered to estimate the smartphone-camera fingerprints.

### Experiments for case study A₁

For this case study, where the disputed images were flat images and a single flat image was used as the reference image to build the smartphone-camera fingerprints, Eq. (6) was used to demonstrate which smartphone-camera captured a disputed flat image. The following seven steps were used to conduct the experiments for this case study.

i. **Evaluating Eq. (6) for a disputed image assuming its own smartphone-camera.** Equation (6) was evaluated under the assumption that the disputed image, $d_{41,01}$, was captured by the smartphone-camera $C_{01}$, *i.e.*, its own capture source. Thus, Table 2 shows the obtained results when the MD between the camera-in-image traces in $d_{41,01}$ and the fingerprint of $C_{01}$ is compared to the MD distance between the camera-in-image traces in $d_{41,01}$ and the fingerprints from the remaining eight smartphone-cameras. Note in Table 2 that the second column represents instances when $MD(T_{41,01},\ F_{01}^1) \leq MD(T_{41,01},\ F_j^1)$ was satisfied, and the third column represents instances when it was not satisfied. Note that $MD(T_{41,01},\ F_{01}^1) \leq MD(T_{41,01},\ F_j^1)$ was always satisfied. This result provides 100% certainty that $d_{41,01}$ was captured by $C_{01}$. Note also that specifically for $MD(T_{41,01},\ F_{01}^1) \leq MD(T_{41,01},\ F_{07}^1)$ both inequality options have a similar number of instances, which differ by about 0.7%. This is because $C_{01}$ and $C_{07}$ are the same brand and model: iPhone SE 2020.

ii. **Evaluating Eq. (6) for a disputed image assuming a different smartphone-camera as the capture source.** As a contrast, Eq. (6) was evaluated under the assumption that $d_{41,01}$ was captured by $C_{02}$. Then, Table 3 shows when $MD(T_{41,01},\ F_{02}^1) \leq MD(T_{41,01},\ F_j^1)$ was satisfied (blue) and when is not (red). Note that in only three out of seven cases $MD(T_{41,01},\ F_{02}^1) \leq MD(T_{41,01},\ F_j^1)$ was satisfied. With this result, it can be assumed that $d_{41,01}$ was captured by $C_{02}$ with a confidence of 42.86%.

iii. **Evaluating Eq. (6) for a disputed image assuming any smartphone-camera as the capture source.** Equation (6) was evaluated under the assumption that $d_{41,01}$ was captured by any smartphone-camera. This assumption generalizes the results of Tables 2 and 3, which are summarized in Table 4, showing that $d_{41,01}$ is associated with $C_{01}$ as the best option (100%) and with $C_{05}$ as the worst option (0%). Note that $C_{07}$ is the second-best option (85.71%), which is congruent since $C_{01}$ and $C_{07}$ are of the same brand and model.

iv. **Evaluating Eq. (6) for the first disputed image from all smartphone-cameras.** Generalizing the previous scenario by considering the first disputed image from all smartphone-cameras, Eq. (6) was evaluated under the assumption that $d_{41,j}$ was captured by any camera. Thus, Table 5 shows the membership results for each $d_{41,j}$ with $j = 1, 2, 3,…, 8$. Note that the first row corresponds to the results shown in Table 4. At this point and considering only one disputed image per smartphone-camera, note in

**Table 2 Comparison results when $d_{41,01}$ is assumed to be an image captured by $C_{01}$ and one flat image was used to estimate the camera fingerprint for each smartphone.**

| Camera fingerprint | $MD(T_{41,01}, F_1^1)$ | $\leq$ | $MD(T_{41,01}, F_j^1)$ |
|---|---|---|---|
| $F_1^1$ | 1,000,000 | | 0 |
| $F_2^1$ | 805,152 | | 194,848 |
| $F_3^1$ | 859,378 | | 140,622 |
| $F_4^1$ | 678,291 | | 321,709 |
| $F_5^1$ | 999,985 | | 15 |
| $F_6^1$ | 680,144 | | 319,856 |
| $F_7^1$ | 501,851 | | 498,149 |
| $F_8^1$ | 995,705 | | 4,295 |

**Table 3 Comparison results when $d_{41,01}$ is assumed to be an image captured by $C_{02}$ and one flat image was used to estimate the camera fingerprint for each smartphone.**

| Camera fingerprint | $MD(T_{41,01}, F_2^1)$ | $\leq$ | $MD(T_{41,01}, F_j^1)$ |
|---|---|---|---|
| $F_1^1$ | 194,848 | | 805,152 |
| $F_2^1$ | 1,000,000 | | 0 |
| $F_3^1$ | 688,940 | | 311,060 |
| $F_4^1$ | 368,670 | | 631,330 |
| $F_5^1$ | 999,989 | | 11 |
| $F_6^1$ | 267,404 | | 732,596 |
| $F_7^1$ | 199,745 | | 800,255 |
| $F_8^1$ | 998,452 | | 1,548 |

**Table 4 Membership of $d_{41,01}$ to each of the eight smartphone-cameras.**

| Assumed membership | Membership (%) | Inequality used |
|---|---|---|
| $d_{41,01}/C_{01}$ | 100.00 | $MD(T_{41,01}, F_1^1) \leq MD(T_{41,01}, F_j^1)$ |
| $d_{41,01}/C_{02}$ | 42.86 | $MD(T_{41,01}, F_2^1) \leq MD(T_{41,01}, F_j^1)$ |
| $d_{41,01}/C_{03}$ | 28.57 | $MD(T_{41,01}, F_3^1) \leq MD(T_{41,01}, F_j^1)$ |
| $d_{41,01}/C_{04}$ | 71.43 | $MD(T_{41,01}, F_4^1) \leq MD(T_{41,01}, F_j^1)$ |
| $d_{41,01}/C_{05}$ | 0.00 | $MD(T_{41,01}, F_5^1) \leq MD(T_{41,01}, F_j^1)$ |
| $d_{41,01}/C_{06}$ | 57.14 | $MD(T_{41,01}, F_6^1) \leq MD(T_{41,01}, F_j^1)$ |
| $d_{41,01}/C_{07}$ | 85.71 | $MD(T_{41,01}, F_7^1) \leq MD(T_{41,01}, F_j^1)$ |
| $d_{41,01}/C_{08}$ | 14.29 | $MD(T_{41,01}, F_8^1) \leq MD(T_{41,01}, F_j^1)$ |

**Table 5 Membership of $d_{41,j}$ assuming they were taken with each of the eight smartphone-cameras.**

| Disputed image | Membership (%) | | | | | | | |
|---|---|---|---|---|---|---|---|---|
| | $C_{01}$ | $C_{02}$ | $C_{03}$ | $C_{04}$ | $C_{05}$ | $C_{06}$ | $C_{07}$ | $C_{08}$ |
| $d_{41,01}$ | 100.00 | 42.86 | 28.57 | 71.43 | 0.00 | 57.14 | 85.71 | 14.29 |
| $d_{41,02}$ | 57.14 | 100.00 | 71.43 | 28.57 | 0.00 | 85.71 | 42.86 | 14.29 |
| $d_{41,03}$ | 57.14 | 100.00 | 71.43 | 28.57 | 0.00 | 85.71 | 42.86 | 14.29 |
| $d_{41,04}$ | 71.43 | 28.57 | 42.86 | 100.00 | 0.00 | 57.14 | 85.71 | 14.29 |
| $d_{41,05}$ | 71.43 | 14.29 | 42.86 | 0.00 | 100.00 | 57.14 | 85.71 | 28.57 |
| $d_{41,06}$ | 57.14 | 85.71 | 71.43 | 28.57 | 0.00 | 100.00 | 42.86 | 14.29 |
| $d_{41,07}$ | 85.71 | 42.86 | 28.57 | 71.43 | 0.00 | 57.14 | 100.00 | 14.29 |
| $d_{41,08}$ | 85.71 | 28.57 | 42.86 | 14.29 | 0.00 | 71.43 | 57.14 | 100.00 |

**Table 6 Membership of $d_{41,j}$ to each of eight smartphone camera, when compared *vs.* their own capture source and one flat image was used to estimate the fingerprint for each smartphone-camera.**

| Assumed membership | Membership (%) | Inequality used |
|---|---|---|
| $d_{41,01}/C_{01}$ | 100.00 | $MD(T_{41,01}, F_1^1) \leq MD(T_{41,01}, F_j^1)$ |
| $d_{41,02}/C_{02}$ | 100.00 | $MD(T_{41,02}, F_2^1) \leq MD(T_{41,01}, F_j^1)$ |
| $d_{41,03}/C_{03}$ | 71.43 | $MD(T_{41,03}, F_3^1) \leq MD(T_{41,01}, F_j^1)$ |
| $d_{41,04}/C_{04}$ | 100.00 | $MD(T_{41,04}, F_4^1) \leq MD(T_{41,01}, F_j^1)$ |
| $d_{41,05}/C_{05}$ | 100.00 | $MD(T_{41,05}, F_5^1) \leq MD(T_{41,01}, F_j^1)$ |
| $d_{41,06}/C_{06}$ | 100.00 | $MD(T_{41,06}, F_6^1) \leq MD(T_{41,01}, F_j^1)$ |
| $d_{41,07}/C_{07}$ | 100.00 | $MD(T_{41,07}, F_7^1) \leq MD(T_{41,01}, F_j^1)$ |
| $d_{41,08}/C_{08}$ | 100.00 | $MD(T_{41,08}, F_8^1) \leq MD(T_{41,01}, F_j^1)$ |
| % Average | 96.43 | |

the diagonal of Table 5 that the effectiveness for the proposed method is 96.43%, considering one flat image to estimate the fingerprint for each smartphone-camera.

v. **Evaluation summary for the first disputed image from all smartphone-cameras.** Therefore, a summary of previous results is shown in Table 6, considering only the diagonal of Table 5. Note that the smallest membership was obtained for $d_{41,03}$ when it was assumed that $C_{03}$ was its capture source, which is a mistake because the proposed method suggests that $d_{41,03}$ was captured by $C_{02}$ (membership of 100%) instead of $C_{03}$ (membership of 71.43%).

vi. **Evaluation summary for all disputed images from all smartphone-cameras.** Generalizing Table 6 for ten disputed flat images, $d_{41,j}$ to $d_{50,j}$ of all smartphone-cameras, and obtaining the diagonal of each of them, Table 7 was prepared. Note that the first row corresponds to the result of the disputed flat images 41, as shown in Table 6. Each column of results of Table 7 shows the membership of the disputed

**Table 7 Membership of $d_{41,j}$ to $d_{50,j}$ to each smartphone-camera assuming one flat image to estimate the smartphone-camera fingerprints.**

| Disputed image | $C_{01}$ | $C_{02}$ | $C_{03}$ | $C_{04}$ | $C_{05}$ | $C_{06}$ | $C_{07}$ | $C_{08}$ | Average |
|---|---|---|---|---|---|---|---|---|---|
| $d_{41,j}$ | 100.00 | 100.00 | 71.43 | 100.00 | 100.00 | 100.00 | 100.00 | 100.00 | 96.43 |
| $d_{42,j}$ | 100.00 | 100.00 | 71.43 | 100.00 | 100.00 | 100.00 | 100.00 | 100.00 | 96.43 |
| $d_{43,j}$ | 100.00 | 100.00 | 71.43 | 100.00 | 100.00 | 100.00 | 100.00 | 100.00 | 96.43 |
| $d_{44,j}$ | 100.00 | 100.00 | 71.43 | 100.00 | 100.00 | 100.00 | 100.00 | 100.00 | 96.43 |
| $d_{45,j}$ | 100.00 | 100.00 | 71.43 | 100.00 | 100.00 | 100.00 | 100.00 | 100.00 | 96.43 |
| $d_{46,j}$ | 100.00 | 100.00 | 71.43 | 100.00 | 100.00 | 100.00 | 100.00 | 100.00 | 96.43 |
| $d_{47,j}$ | 100.00 | 100.00 | 71.43 | 100.00 | 100.00 | 100.00 | 100.00 | 100.00 | 96.43 |
| $d_{48,j}$ | 100.00 | 100.00 | 71.43 | 100.00 | 100.00 | 100.00 | 100.00 | 100.00 | 96.43 |
| $d_{49,j}$ | 100.00 | 100.00 | 71.43 | 100.00 | 100.00 | 100.00 | 100.00 | 100.00 | 96.43 |
| $d_{50,j}$ | 100.00 | 100.00 | 71.43 | 100.00 | 100.00 | 100.00 | 100.00 | 100.00 | 96.43 |
| %Average | 100.00 | 100.00 | 71.43 | 100.00 | 100.00 | 100.00 | 100.00 | 100.00 | 96.43 |

**Table 8 Effectiveness of the proposed method when one flat image was used to estimate the smartphone-camera fingerprints.**

| Disputed image | $C_{01}$ | $C_{02}$ | $C_{03}$ | $C_{04}$ | $C_{05}$ | $C_{06}$ | $C_{07}$ | $C_{08}$ |
|---|---|---|---|---|---|---|---|---|
| $d_{41,j}$ | $C_{01}$ | $C_{02}$ | $C_{02}$ | $C_{04}$ | $C_{05}$ | $C_{06}$ | $C_{07}$ | $C_{08}$ |
| $d_{42,j}$ | $C_{01}$ | $C_{02}$ | $C_{02}$ | $C_{04}$ | $C_{05}$ | $C_{06}$ | $C_{07}$ | $C_{08}$ |
| $d_{43,j}$ | $C_{01}$ | $C_{02}$ | $C_{02}$ | $C_{04}$ | $C_{05}$ | $C_{06}$ | $C_{07}$ | $C_{08}$ |
| $d_{44,j}$ | $C_{01}$ | $C_{02}$ | $C_{02}$ | $C_{04}$ | $C_{05}$ | $C_{06}$ | $C_{07}$ | $C_{08}$ |
| $d_{45,j}$ | $C_{01}$ | $C_{02}$ | $C_{02}$ | $C_{04}$ | $C_{05}$ | $C_{06}$ | $C_{07}$ | $C_{08}$ |
| $d_{46,j}$ | $C_{01}$ | $C_{02}$ | $C_{02}$ | $C_{04}$ | $C_{05}$ | $C_{06}$ | $C_{07}$ | $C_{08}$ |
| $d_{47,j}$ | $C_{01}$ | $C_{02}$ | $C_{02}$ | $C_{04}$ | $C_{05}$ | $C_{06}$ | $C_{07}$ | $C_{08}$ |
| $d_{48,j}$ | $C_{01}$ | $C_{02}$ | $C_{02}$ | $C_{04}$ | $C_{05}$ | $C_{06}$ | $C_{07}$ | $C_{08}$ |
| $d_{49,j}$ | $C_{01}$ | $C_{02}$ | $C_{02}$ | $C_{04}$ | $C_{05}$ | $C_{06}$ | $C_{07}$ | $C_{08}$ |
| $d_{50,j}$ | $C_{01}$ | $C_{02}$ | $C_{02}$ | $C_{04}$ | $C_{05}$ | $C_{06}$ | $C_{07}$ | $C_{08}$ |
| **Effectiveness rate: 85.7%** | 100% | 100% | **0%** | 100% | 100% | 100% | 100% | 100% |

images to the same smartphone-camera when it was assumed that they were captured by their own capture device.

vii. **Estimating the effectiveness of the proposed method for disputed flat images.** Table 7 shows that the membership of the proposed method for ten flat disputed images is 96.43% when one flat image was used to estimate the fingerprint of each smartphone-camera. Note that the flat disputed images from $C_{03}$ generated a membership of 71.43% for their own smartphone-camera, as the proposed method assigned them to $C_{02}$ instead of $C_{03}$ (see Table 5). Thus, according to the results shown in Table 8, the effectiveness rate of the proposed method was 85.70% considering that

the proposed method failed for the disputed images from $C_{03}$, by assigning them to $C_{02}$ instead $C_{03}$.

### Experiments for case study $A_2$

For this case study, where the disputed images are flat and fifteen reference flat images were used to build the smartphone-camera fingerprints, Eq. (6) was used to demonstrate which smartphone-camera was the capture source of each disputed flat image. In this case study, the proposed method was modified to improve its effectiveness rate. Therefore, instead of one reference flat image, fifteen flat images were considered. Following the seven steps of the previous case, the obtained results of the membership for the disputed images numbered from $d_{41,j}$ to $d_{50,j}$ are summarized in Table 9. In addition, Table 10 shows that the effectiveness rate of the proposed method turned out to be 100%. It should be noted that, using fifteen reference flat images instead one to estimate the smartphone-camera fingerprints, the effectiveness rate of the proposed method reached 100%.

### Experiments for case study B

For the experiments in this case study, ten natural images were selected as disputed images, which corresponds to more realistic case, since it includes natural images that containing scenes with objects, animals, buildings, and views. According to description of proposed method and considering the results obtained of the experiments for case study $A_2$, fifteen flat images were used to estimate the smartphone-camera fingerprints. It is important to specify that the disputed natural images were processed to obtain from each one of them a synthetic flat image (flat equivalent image) that uniquely corresponds to its origin. Thus, in the proposed method, each synthetic flat image was considered as the disputed image used to determine the capture source of its corresponding natural image. Remember that each flat synthetic image must be created by selecting from a disputed natural image the pixels whose intensity is within the specified selection interval. Following the seven steps described in the "Experiments for case study $A_1$" section, the average membership for the flat synthetic images numbered from $d_{41,j}$ to $d_{50,j}$ was 98.90% and is summarized in Table 11. In addition, Table 12 shows that the effectiveness rate of the proposed method was 97.5%.

Note in Table 11 that, although there are events that achieve a membership of 100% with their capture source, the proposed method does not reach an effectiveness rate of 100%, as it fails when considering natural images from smartphone-cameras $C_{01}$, $C_{02}$, $C_{03}$, $C_{06}$, and $C_{07}$. In Table 11, these events are highlighted with copper. In addition, Table 12 details the confusion situations. Relating Tables 11 to 12, it can be noted that when the disputed natural images $d_{42,j}$ were assumed from $C_{01}$, they reached a membership of 100% with $C_{01}$ and $C_{07}$. However, when the disputed natural images $d_{50,j}$ were assumed from $C_{01}$, they reached a membership of 85.70% also with $C_{01}$ and $C_{07}$. On the other hand, when the disputed natural images $d_{42,j}$ were assumed from $C_{02}$, the proposed method estimated a membership of 85.70% with $C_{02}$, but 100% with $C_{03}$. Additionally, when the disputed natural images $d_{48,j}$ were assumed from $C_{03}$, the proposed method estimated a

**Table 9 Membership of the disputed images from $d_{41,j}$ to $d_{50,j}$ assuming fifteen reference images to estimate the smartphone-camera fingerprints**

| Disputed image | $C_{01}$ | $C_{02}$ | $C_{03}$ | $C_{04}$ | $C_{05}$ | $C_{06}$ | $C_{07}$ | $C_{08}$ | Average |
|---|---|---|---|---|---|---|---|---|---|
| $d_{41,j}$ | 100 | 100 | 100 | 100 | 100 | 100 | 100 | 100 | 100 |
| $d_{42,j}$ | 100 | 100 | 100 | 100 | 100 | 100 | 100 | 100 | 100 |
| $d_{43,j}$ | 100 | 100 | 100 | 100 | 100 | 100 | 100 | 100 | 100 |
| $d_{44,j}$ | 100 | 100 | 100 | 100 | 100 | 100 | 100 | 100 | 100 |
| $d_{45,j}$ | 100 | 100 | 100 | 100 | 100 | 100 | 100 | 100 | 100 |
| $d_{46,j}$ | 100 | 100 | 100 | 100 | 100 | 100 | 100 | 100 | 100 |
| $d_{47,j}$ | 100 | 100 | 100 | 100 | 100 | 100 | 100 | 100 | 100 |
| $d_{48,j}$ | 100 | 100 | 100 | 100 | 100 | 100 | 100 | 100 | 100 |
| $d_{49,j}$ | 100 | 100 | 100 | 100 | 100 | 100 | 100 | 100 | 100 |
| $d_{50,j}$ | 100 | 100 | 100 | 100 | 100 | 100 | 100 | 100 | 100 |
| **%Average** | 100 | 100 | 100 | 100 | 100 | 100 | 100 | 100 | 100 |

**Table 10 Effectiveness of the proposed method when fifteen flat images are used to estimate the smartphone-camera fingerprints.**

| Disputed image | $C_{01}$ | $C_{02}$ | $C_{03}$ | $C_{04}$ | $C_{05}$ | $C_{06}$ | $C_{07}$ | $C_{08}$ |
|---|---|---|---|---|---|---|---|---|
| $d_{41,j}$ | $C_{01}$ | $C_{02}$ | $C_{03}$ | $C_{04}$ | $C_{05}$ | $C_{06}$ | $C_{07}$ | $C_{08}$ |
| $d_{42,j}$ | $C_{01}$ | $C_{02}$ | $C_{03}$ | $C_{04}$ | $C_{05}$ | $C_{06}$ | $C_{07}$ | $C_{08}$ |
| $d_{43,j}$ | $C_{01}$ | $C_{02}$ | $C_{03}$ | $C_{04}$ | $C_{05}$ | $C_{06}$ | $C_{07}$ | $C_{08}$ |
| $d_{44,j}$ | $C_{01}$ | $C_{02}$ | $C_{03}$ | $C_{04}$ | $C_{05}$ | $C_{06}$ | $C_{07}$ | $C_{08}$ |
| $d_{45,j}$ | $C_{01}$ | $C_{02}$ | $C_{03}$ | $C_{04}$ | $C_{05}$ | $C_{06}$ | $C_{07}$ | $C_{08}$ |
| $d_{46,j}$ | $C_{01}$ | $C_{02}$ | $C_{03}$ | $C_{04}$ | $C_{05}$ | $C_{06}$ | $C_{07}$ | $C_{08}$ |
| $d_{47,j}$ | $C_{01}$ | $C_{02}$ | $C_{03}$ | $C_{04}$ | $C_{05}$ | $C_{06}$ | $C_{07}$ | $C_{08}$ |
| $d_{48,j}$ | $C_{01}$ | $C_{02}$ | $C_{03}$ | $C_{04}$ | $C_{05}$ | $C_{06}$ | $C_{07}$ | $C_{08}$ |
| $d_{49,j}$ | $C_{01}$ | $C_{02}$ | $C_{03}$ | $C_{04}$ | $C_{05}$ | $C_{06}$ | $C_{07}$ | $C_{08}$ |
| $d_{50,j}$ | $C_{01}$ | $C_{02}$ | $C_{03}$ | $C_{04}$ | $C_{05}$ | $C_{06}$ | $C_{07}$ | $C_{08}$ |
| **Effectiveness** | 100% | 100% | 100% | 100% | 100% | 100% | 100% | 100% |

membership of 85.70% with smartphone-cameras $C_{02}$ and $C_{03}$. Note that when the disputed natural images $d_{50,j}$ were assumed from $C_{06}$, the proposed method estimated a membership of 71.40% with $C_{06}$, but 100% with $C_{03}$. Finally, when the disputed natural images $d_{47,j}$ were assumed from $C_{07}$, the proposed method estimated a membership of 100.00% with $C_{01}$ and $C_{07}$.

# DISCUSSION AND COMPARISON WITH OTHER METHODS

Among the studies discussed in the "Related Works" section, it is worth noting those that propose source camera identification methods that involve statistical analysis. They defined the camera fingerprints from the statistical distribution of the PRNU extracted from a set of reference flat images. In these methods, a divergence measure was computed

**Table 11 Membership of the disputed images from $d_{41}$ to $d_{50}$ for each smartphone-camera assuming fifteen flat images to estimate the smartphone-camera fingerprints.**

| Disputed image | $C_{01}$ | $C_{02}$ | $C_{03}$ | $C_{04}$ | $C_{05}$ | $C_{06}$ | $C_{07}$ | $C_{08}$ | Average |
|---|---|---|---|---|---|---|---|---|---|
| $d_{41,j}$ | 100.0 | 100.0 | 100.0 | 100.0 | 100.0 | 100.0 | 100.0 | 100.0 | 100.0 |
| $d_{42,j}$ | 100.0 | 85.7 | 100.0 | 100.0 | 100.0 | 100.0 | 100.0 | 100.0 | 98.2 |
| $d_{43,j}$ | 100.0 | 100.0 | 100.0 | 100.0 | 100.0 | 100.0 | 100.0 | 100.0 | 100.0 |
| $d_{44,j}$ | 100.0 | 100.0 | 100.0 | 100.0 | 100.0 | 100.0 | 100.0 | 100.0 | 100.0 |
| $d_{45,j}$ | 100.0 | 100.0 | 100.0 | 100.0 | 100.0 | 100.0 | 100.0 | 100.0 | 100.0 |
| $d_{46,j}$ | 100.0 | 100.0 | 100.0 | 100.0 | 100.0 | 100.0 | 100.0 | 100.0 | 100.0 |
| $d_{47,j}$ | 100.0 | 100.0 | 100.0 | 100.0 | 100.0 | 100.0 | 85.7 | 100.0 | 98.2 |
| $d_{48,j}$ | 100.0 | 100.0 | 85.7 | 100.0 | 100.0 | 100.0 | 100.0 | 100.0 | 98.2 |
| $d_{49,j}$ | 100.0 | 100.0 | 100.0 | 100.0 | 100.0 | 100.0 | 100.0 | 100.0 | 100.0 |
| $d_{50,j}$ | 85.7 | 100.0 | 100.0 | 100.0 | 100.0 | 71.4 | 100.0 | 100.0 | 94.6 |
| %Average | 98.60 | 98.60 | 98.60 | 100.00 | 100.00 | 97.10 | 98.70 | 100.00 | 98.90 |

**Table 12 Effectiveness of the proposed method when fifteen reference images were used to obtain the smartphone-camera fingerprints.**

| Disputed image | $/C_{01}$ | $/C_{02}$ | $/C_{03}$ | $/C_{04}$ | $/C_{05}$ | $/C_{06}$ | $/C_{07}$ | $/C_{08}$ |
|---|---|---|---|---|---|---|---|---|
| $d_{41,j}$ | $C_{01}$ | $C_{02}$ | $C_{03}$ | $C_{04}$ | $C_{05}$ | $C_{06}$ | $C_{07}$ | $C_{08}$ |
| $d_{42,j}$ | $C_{01}, C_{07}$ | $C_{03}$ | $C_{03}$ | $C_{04}$ | $C_{05}$ | $C_{06}$ | $C_{07}$ | $C_{08}$ |
| $d_{43,j}$ | $C_{01}$ | $C_{02}$ | $C_{03}$ | $C_{04}$ | $C_{05}$ | $C_{06}$ | $C_{07}$ | $C_{08}$ |
| $d_{44,j}$ | $C_{01}$ | $C_{02}$ | $C_{03}$ | $C_{04}$ | $C_{05}$ | $C_{06}$ | $C_{07}$ | $C_{08}$ |
| $d_{45,j}$ | $C_{01}$ | $C_{02}$ | $C_{03}$ | $C_{04}$ | $C_{05}$ | $C_{06}$ | $C_{07}$ | $C_{08}$ |
| $d_{46,j}$ | $C_{01}$ | $C_{02}$ | $C_{03}$ | $C_{04}$ | $C_{05}$ | $C_{06}$ | $C_{07}$ | $C_{08}$ |
| $d_{47,j}$ | $C_{01}$ | $C_{02}$ | $C_{03}$ | $C_{04}$ | $C_{05}$ | $C_{06}$ | $C_{01}, C_{07}$ | $C_{08}$ |
| $d_{48,j}$ | $C_{01}$ | $C_{02}$ | $C_{02}, C_{03}$ | $C_{04}$ | $C_{05}$ | $C_{06}$ | $C_{07}$ | $C_{08}$ |
| $d_{49,j}$ | $C_{01}$ | $C_{02}$ | $C_{03}$ | $C_{04}$ | $C_{05}$ | $C_{06}$ | $C_{07}$ | $C_{08}$ |
| $d_{50,j}$ | $C_{01}, C_{07}$ | $C_{02}$ | $C_{03}$ | $C_{04}$ | $C_{05}$ | $C_{03}$ | $C_{07}$ | $C_{08}$ |
| Effectiveness | 100% | 90% | 100% | 100% | 100% | 90% | 100% | 100% |

between each camera fingerprint and the camera-in-image traces left by the digital camera in each captured image. For instance, to solve the problem of identifying the capture source from digital images, *Long, Peng & Zhu (2019)* developed a camera identification method based on the binary Kullback-Leibler divergence (KLD) and an SVM classifier for 36-dimensional features and *Quintanar-Reséndiz et al. (2021)* developed an one-dimensional statistical classifier based on the KLD. In addition, *Quintanar-Reséndiz et al. (2022)* developed another one-dimensional classifier based on the Hellinger distance (HLD) and *Rodríguez-Santos et al. (2022)* developed another based on the Jensen-Shannon divergence (JSD). It is important to mention that *Long, Peng & Zhu, 2019* used natural images (DNI) and computer generated images (CGI) as disputed images and *Rodríguez-Santos et al.*

*(2022)* used disputed flat images (DFI) and DNI. However, *Quintanar-Reséndiz et al. (2021, 2022)* used only DFI. On the other hand, the method proposed by *Long, Peng & Zhu (2019)* used whole reference flat images to compute statistical camera fingerprints. Whereas the methods proposed by *Quintanar-Reséndiz et al. (2021)*, *Rodríguez-Santos et al. (2022)*, and *Quintanar-Reséndiz et al. (2022)* used clippings of the reference flat images for the same purpose. Notably, *Quintanar-Reséndiz et al. (2021)* found that a clipping size greater than or equal to $256 \times 256$ extracted from reference flat images was sufficient to obtain the best identification rate in their method. However, they decided to use a clipping size of $500 \times 500$. This consideration also was adopted by *Quintanar-Reséndiz et al. (2022)*. Furthermore, based on a behavioral analysis of the statistical distribution of the PRNU, *Quintanar-Reséndiz et al. (2021)* also determined the number of reference flat images that should be used to estimate the camera fingerprints. That is, using the mean square error (MSE) between the average statistical distributions of the PRNU drawn from $t$ and $t - 1$ reference images, they determined that 20 reference images were sufficient to obtain an invariant statistical distribution to represent the camera fingerprints. However, they decided to use 30 reference flat images to estimate the camera fingerprints, achieving an identification rate of 99.35% using DFI. Moreover, *Rodríguez-Santos et al. (2022)* defined two case studies to identify the source camera. In one case, they used 20 DFI and in the other case they used 20 DNI. They further constrained the statistical distributions of the PRNU to have a width of $(2 \times \delta)$ with $\delta = 8$, $\delta = \sigma$, and $\delta = 2$. When performing JSD-based comparisons, they computed these distributions considering the same number of intervals in the statistical partition. They reported that the identification rate was 90.40% with DFI but turned out to be about 75% with DNI. In contrast, the method proposed in this study achieved an identification rate of 87.50% for identifying the source camera from DFI when a single reference flat image was used to estimate the camera fingerprints. However, the identification rate increases to 100.00% when 15 reference flat images were used to estimate the camera fingerprints. In this case, remember that the reference flat images had a clipping of $1,000 \times 1,000$. On the other hand, for the case study with DNI, where a gray pixel selection technique was previously applied, from which the PRNU was extracted to determine the camera-in-image traces left by the smartphone-camera on the DNI, the proposed method achieved an identification rate of 97.50% when 15 flat reference images were used. The features for each camera identification method are summarized in Table 13 including the proposed method in this study. Besides the above works, the methods proposed by *Roy et al. (2017)* and *Behare, Bhalchandra & Kumar (2019)* were included for comparison. Moreover, *Roy et al. (2017)* proposed a method based on camera fingerprints computed by extracting the DCT residual features and using a random forest algorithm with AdaBoost for classification. Meanwhile, *Behare, Bhalchandra & Kumar (2019)* proposed a PRNU-based method that compares one-dimensional features using correlation functions. Note that Table 13 shows the image dataset used in each case. The Columbia University image dataset was developed by *Ng et al. (2004)* and is available at https://www.ee.columbia.edu/ln/dvmm/downloads/PIM_PRCG_dataset/, the Dresden image dataset was developed by *Gloe & Böhme (2010)* and is available at https://www.kaggle.com/datasets/micscodes/dresden-image-database,

**Table 13 Comparison of the PRNU-based methods for the identification of the capture device.**

| Method | Image dataset | Images | Devices | Identification strategy | Identification rate |
|--------|--------------|--------|---------|------------------------|---------------------|
| *Roy et al. (2017)* | Dresden | 10,507 | 10 | Camera fingerprints were calculated by extracting the DCT residual features and using a subsequent Random Forest based ensemble classification with AdaBoost. | 99.10% |
| *Long, Peng & Zhu (2019)* | Columbia-University, Dresden, and Internet | 2,000 3,400 600 | Not reported | PRNU-based camera fingerprints from the RGB image channels and binary KLD, binary histogram distance (minimum and absolute), and binary mutual entropy computed to obtain 36-dimensional features comparable using SVM. | 99.83% (average): 99.91% (DNI) and 99.75% (CGI). |
| *Behare, Bhalchandra & Kumar (2019)* | Dresden | 125 | 25 | PRNU-based camera fingerprints from individual channels in RGB images compared using correlation functions. | 100.00%. |
| *Quintanar-Reséndiz et al. (2021)* | Dresden | 400 | 8 | PRNU-based statistical camera fingerprints from the green channel in RGB images, computed to obtain one-dimensional features, which were compared using KLD. | 99.35% (DFI). |
| *Quintanar-Reséndiz et al. (2022)* | Dresden | 400 | 8 | PRNU-based statistical camera fingerprints from the green channel in RGB images, computed to obtain one-dimensional features, which were compared using HLD. | 97.68% (DFI). |
| *Rodríguez-Santos et al. (2022)* | HDRI | 520 | 13 | PRNU-based statistical camera fingerprints from the green channel in RGB images, computed to obtain one-dimensional features, which were compared using JSD. | 90.40% (DFI). 73.46% (DNI, $\delta = 8$). 75.38% (DNI, $\delta = \sigma$). 73.85% (DNI, $\delta = 2$). |
| Proposed method | IPN-NFID | 400 | 8 | Camera fingerprints based on the PRNU and pixel intensity from the green channel in RGB images, computed to obtain two-dimensional features that were compared using MD-based discriminant analysis. | 87.50% (DFI, one reference image). 100.00% (DFI, 15 reference images). 97.50% (DNI, 15 reference images). |

HDRI dataset was developed by *Shaya et al. (2018)* in the Communications and Signal Processing Laboratory of the University of Florence in Italy and it is available at https://lesc.dinfo.unifi.it/en/datasets, and the IPN-NFID image dataset was developed by *Rojas-López et al. (2024)* and it is available at https://doi.org/10.6084/m9.figshare.25201319.

More experiments could be done with more smartphone-cameras, including a wider variety of brands and models. This would allow to confirm that the proposed method has a wide coverage of digital camera brands and models, and it is not only applicable to smartphone-cameras. However, this action is not necessary to guarantee the effectiveness of the proposed method, since as analyzed by *Quintanar-Reséndiz et al. (2021)*, 20 reference images per camera is an acceptable number for capture source identification when using one-dimensional variables. In addition, in a real forensic case a reduced number of digital cameras to be analyzed are involved in the identification of the capture source. This statement is confirmed by the results reported by the authors listed in Table 13. With this baseline in mind, and assuming two-dimensional variables, in this

study 15 reference images were used to build smartphone-camera fingerprints, with consistent results across experiments.

## CONCLUSIONS

Most of the methods and algorithms that have been reported to date for identifying smartphone-cameras from digital images are primarily based on PRNU. The main reason for the extensive use of PRNU is that it is considered a stochastic, inevitable, universal, permanent, and robust signal that the cameras leave in the digital images captured by them. Assuming that in a digital image the PRNU is related to the pixel intensity, this study presented as a novelty that it is possible to use both variables to constitute a two-dimensional variable that can provide more information about the camera-in-image traces. Thus, this experimental condition can be exploited through a two-degree-of-freedom discriminant analysis based on the Mahalanobis distance to reduce the number of reference digital images required to build the camera fingerprints. This study demonstrates that applying a two-degree-of-freedom Mahalanobis classifier, the average effectiveness of the proposed method was 97.5% when the disputed images were natural images, and $1,000 \times 1,000$ clippings were extracted from the reference flat images used to compute the smartphone-camera fingerprints. On the other hand, it should be noted that one area not addressed in this study was the examination of other filtering alternatives to extract the PRNU. It is possible that other filtering techniques, such as the Mihcak denoising filter, could help to improve the identification efficiency of the proposed method.

## ACKNOWLEDGEMENTS

The authors thank A. L. Quintanar-Reséndiz and R. F. Santana-Cruz for the technical assistance and advice in the experimental setup of this work.

### Funding

This work was supported by the Instituto Politécnico Nacional (IPN-México) through projects SIP–20240745 and SIP-20242843 (Rubén Vázquez–Medina), and SIP–20240749 (Omar Jiménez-Ramírez). The funders had no role in study design, data collection and analysis, decision to publish, or preparation of the manuscript.

### Grant Disclosures

The following grant information was disclosed by the authors:
Instituto Politécnico Nacional (IPN-México): SIP–20240745, SIP-20242843, and SIP–20240749.

### Competing Interests

The authors declare that they have no competing interests.

## Author Contributions

- Rubén Vázquez-Medina conceived and designed the experiments, analyzed the data, performed the computation work, prepared figures and/or tables, authored or reviewed drafts of the article, and approved the final draft.
- César Enrique Rojas-López performed the experiments, performed the computation work, prepared figures and/or tables, authored or reviewed drafts of the article, and approved the final draft.
- Omar Jiménez-Ramírez performed the experiments, prepared figures and/or tables, and approved the final draft.
- Luis Niño-de-Rvera-Oyarzabal analyzed the data, authored or reviewed drafts of the article, and approved the final draft.
- Leonardo Palacios-Luengas analyzed the data, authored or reviewed drafts of the article, and approved the final draft.

## Data Availability

The raw data and code for IPN-NFID-Mahalanobis is available at figshare: Rojas López, César Enrique (2024). Raw data and code for IPN-NFID-Mahalanobis. figshare. Dataset. https://doi.org/10.6084/m9.figshare.25153358.v6.

The IPN-NFID Dataset is available at figshare: Rojas López, César Enrique (2024). IPN-NFID Dataset. figshare. Dataset. https://doi.org/10.6084/m9.figshare.25201319.v2.

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
