# Peer review of "Two-degree of freedom Mahalanobis classifier for smartphone-camera identification from natural digital images"

_PeerJ Computer Science, doi:10.7717/peerj-cs.2513_

## Round 0.1 · original submission · Major Revisions

Dear Authors,

The reviewers of your paper have pointed to necessary improvements for acceptance of your manuscript. Among others, following enhancements are mandatory:

/> A solid justification why Mahalanobis is used. And why not another classical coherence method and why not ML found in other works
/> An in-depth comparison with the several other works about camera identification found in literature
/> improve the description and justification of several issues of the experiments,
/> redo your experiments with a larger set of smartphone cameras of more diverse makes and models
/> enhance the quality and the coherence of the figures and images.

We believe that by fixing all the improvements solicited by the reviewers your work will be much improved. I hope that you are able to do them in the next few weeks.

Reviewer 1 ·

Basic reporting

1. The authors should try to keep the indexes more consistent since they seem to refer to different objects in various parts of the work:
line 196, 197,”image i taken by a smartphone camera j”
Line 200 “k flat images taken under controlled conditions by a smartphone camera l”
Line 220 “with j = 1 or j = 2, are n-dimensional features of the class wi, with i = 1,2,3, . . . , p(s j |wi) ”
Line 234 “the k-th reference digital image of the l-th smartphone camera”
Line 240 “i-th disputed flat image of the j-th smartphone camera”
Line 256 “where l is the first index that changes in the comparison”

2. There are several acronyms which are not explained at the first usage in the text, i.e., EEG, RGB, DCT.

3. There is a typo at line 232 “Considering fifteen reference digital images by each one of ten smartphone cameras”. Also there are several minor typos, e.g., line 204, line 235, line 333, etc.

Experimental design

1. A picture (flowchart) with the steps of image processing steps leading to camera identification would help for an overview of the method in this work.

2. The number of cameras and images used in the work is very limited compared with those from existing works. Why only 8 from a total of 12 smartphone cameras available in the IPN-NFID dataset were used for the identification in this work? What about the other 4? Why only 15 and 10 images were used (Table 1 mentions 180 images with each phone)? Would be interesting to see how the proposed method is working on the largest dataset.

Validity of the findings

1. As also the authors mentioned, there are a lot of works which proposed several methods for camera identification. A comparison of the methods, number of devices, number of images and results used in the literature would improve the quality of this work.

2. The authors should add a motivation for the usage of the Mahalanobis distance in the introduction. Why was Mahalanobis used? Why not use a classical coherence which is used in several papers in the literature? Or the Euclidean or Hamming distances? Why not use machine learning algorithms / neural networks?

Additional comments

1. The pdf is very large. The pages with figure 3 and 4 are loading too slow. I suggest reducing the size of the images from these figures.

Reviewer 2 ·

Basic reporting

The article is written in clear English. There are given literature references concerning the subject of the paper. Overall, the structure of the paper is correct.

I think the some of the discussion on the theory should be improved. For example, the chapter on "Intrinsic noise in digital images" is very brief and it requires the reader to look to the references to understand the details. I think that it would be beneficial to desribe briefly, but with necessary details the applied filtering techniques.
It would be easier to understand why PRNU given by Eq.4 has negative values as shown in the figure 4. It is image signal distorted by a noise, according the the formula, so how can it comtain negative values?

Experimental design

I think that the desription of the research should be improved.
For example, it is written in 132 that all shots were taken under the same lighting conditions in a room with a controlled lighting. Was it true also for natural images? It is not clear from the context.
The lighting conditions would have a large effect on the resultant images - not only the overall brightness of pixels will cause a shift of a flat image's histogram left or right, but also the entire noise pattern in the image would change as one can observe by taking a shot in poor light condiotions - the noise will be increased. So how one can compare a disputed image which may be taken under such adverse conditions?
Line 139 - the images clearly do not show the same scene taken by different cameras
Line 143 - Why only green channel was considered in investigation?

Validity of the findings

I think that the data which is used for making a fingeprint of cell phone camera should be desribed in much more detail. Now, the reader may understand that the authors try to compare a reference histogram vs PRNU feature made from flat grey images (as shown in figures 3 and 4) with same feature but made using the natural scene images (i0 suggests that it is image, not noise extracted from the image) - and I think that this approach is impossible to success because the pixel intensity distribution will change when the photographed scene and illumination condition change.
I think it is feasable to classify photos as taken from different cameras if one can extract the noise pattern, at the same time normalizing it so there is no dependence on the lighting conditions (causing shift in histograms)

Additional comments

I think that the paper needs major revision and clarification of issues, some of which I presented. Then it may be considered for further review and publication.

---

## Round 0.2 · accepted · Accept

Since all the requested modifications and improvements have been implemented in this new version, the manuscript is ready for publication. Congratulations.

Reviewer 1 ·

Basic reporting

All complains form the first review were solved. No additional points.

Experimental design

All complains form the first review were solved. No additional points.

Validity of the findings

All complains form the first review were solved. No additional points.

Additional comments

The paper was improved in comparison with the first review.
All complains form the first review were solved. No additional points.